# The Ultra-Large-Bandwidth Cascade Full-Stokes-Imaging Metasurface Based on the Dual-Major-Axis Circular Dichroism Grating

**DOI:** 10.3390/nano13152211

**Published:** 2023-07-30

**Authors:** Bo Cheng, Guofeng Song

**Affiliations:** 1Institute of Semiconductors, Chinese Academy of Sciences, Beijing 100083, China; chengbo9610@semi.ac.cn; 2College of Materials Science and Opto-Electronic Technology, University of Chinese Academy of Sciences, Beijing 100049, China

**Keywords:** full Stokes pixel, circular dichroism, metasurface, FP cavity resonance

## Abstract

A dual-major-axis grating composed of two metal–insulator–metal (MIM) waveguides with different dielectric layer thicknesses is numerically proposed to achieve the function of the quarter-wave plate with an extremely large bandwidth (1.0–2.2 μm), whose optical properties can be controlled by the Fabry–Pérot (FP) resonance. For the TE incident mode wave, MIM waveguides with large (small) dielectric layer thicknesses control the guided-mode resonant channels of long (short) waves, respectively, in this miniaturized optical element. Meanwhile, for the TM incident mode wave, the propagation wave vector of this structure is controlled by the hybrid mode of two gap-SPPs (gap-surface plasmon polaritons) with different gap thicknesses. We combine this structure with a thick silver grating to propose a circularly polarizing dichroism device, whose effective bandwidth can reach an astonishing 1.65 μm with a circular polarization extinction ratio greater than 10 dB. The full Stokes pixel based on the six-image element technique can almost accurately measure arbitrary polarization states at 1.2–2.8 μm (including elliptically polarized light), which is the largest bandwidth (1600 nm) of the full Stokes large-image element to date in the near-infrared band. In addition, the average errors of the degree of linear polarizations (Dolp) and degree of circular polarizations (Docp) are less than −25 dB and −10 dB, respectively.

## 1. Introduction

Much useful information that is difficult to detect directly can be obtained by manipulating the polarization of light, such as surface shape, roughness, and chemical properties [1,2,3]. A metasurface consisting of a two-dimensional array of meta-atoms with a subwavelength period can flexibly and precisely manipulate the phase and polarization information of light [4,5,6]. The polarization camera concepts are mainly divided into three categories: the division of the amplitude [7], division of the aperture [8], and division of the focal plane [9,10,11]. The division of the focal plane based on optical metasurfaces can be naturally extended to imaging chips with ultra-large arrays and ultra-small pixel spacing. In addition, the development of the compact polarization camera is limited mainly by the bandwidth, the extinction ratio, and the circular dichroism of the circular dichroic device. The extinction ratio and the circular dichroism correspond to the most essential polarization perception, and the effective bandwidth affects the strength of the signal in the signal-to-noise ratio of the imaging system.

In the past ten years, circular dichroic devices based on metasurfaces have developed rapidly [12,13,14,15,16,17], and their design concepts are mainly divided into two categories: the chiral geometric structure assembly method [18,19,20,21,22] and wave plate combination method [23,24,25]. The chiral geometric structure assembly method is mainly used to numerically scan a chiral geometric structure cell, such as Archimedean spirals or zigzags, to obtain the circular polarization control effect through the parameter traversal method. In 2007, Fedotov [21] achieved the plasmon excitation coupling of special chiral light by tuning the size of the nanostructured, chiral, twisted-fish-scale metamaterial at the nanoscale. The following year, Schwanecke and Fedotov [20] continued to deeply optimize the two-dimensional, planar, twisted-fish-scale structure, and they developed the first asymmetric transmission in the chiral domain based on the planar photonic metamaterial while ensuring high circular dichroism. In 2013, Chen [18] used the antisymmetric array of spatially triangular apertures to achieve the surface plasma excitation coupling of the azimuthal and radial polarization as well as polarization focusing, with an extinction ratio of about 16:1 under small-pixel detection conditions. In 2013, Bachman [19] investigated the ability of nested Archimedean spiral gratings of different sizes to manipulate circularly polarized light, and the grating was very easy to integrate with detectors and had an extinction ratio of about 6:1, but was severely limited by a low transmission efficiency (about 10%). In 2015, Esposito [16] used the innovative tomographic rotatory growth technique to prepare high-quality, three-dimensional, triple-helical nanowires, and the devices showed up to 37% circular dichroism in a large bandwidth (500–1000 nm), with a 24 dB signal-to-noise ratio. In 2017, Hu [22] used an amorphous silicon metasurface with the cubic unit cell of z-shaped holes to separate left and right circularly polarized light, and the author was able to obtain high-quality, circularly dichroic wave plates with a circular dichroism of up to 90% and extinction ratios close to 10:1 by the large-scale scanning of geometric parameters. In 2022, Esposito [17] proposed a 3D, compact, chiral metacrystal whose chiroptical properties were finely manipulated by in-plane and out-of-plane diffractive coupling, highlighting perspectives on novel schemes of enantiomeric detection. The wave plate combination method mainly draws on the combination of a quarter-wave plate (QWP) and a 45-degree linear polarizer in traditional optics, and some breakthrough achievements have been reported regarding high circular dichroism and high extinction ratios [26,27,28,29,30,31,32]. In 2019, Bai [32] constructed a high-quality circular polarizer using a pair of cross-shaped, ultrathin, plasmonic QWPs and a linearly polarized grating, but the 50% transmittance efficiency limit [33] of the ultrathin metal QWP resulted in the circular dichroism of the entire circular polarizer being less than 20%. In the same year, Basiri [23] used a silicon dielectric QWP without ohmic loss instead of the metal QWP, and the efficiency of the whole circular dichroic wave plate was improved to 80%. However, the effective bandwidth in the shortwave band still did not exceed 400 nm. In addition, there were also some chiral manipulations through quasi-bound states in the continuum (QBICS) [34,35,36,37], but the narrowband effect induced by the mode interference characteristics implies that its application may lean towards sensing rather than detection.

In this article, we use the thick silver grating to design 0-, 45-, 90- and 135-degree polarizers with transmission efficiencies exceeding 90% based on the bandpass and bandstop effects of the waveguide mode. Meanwhile, we utilize a bilayer plasmonic metasurface, a combination of the QWP and linear polarizer, to propose a circularly polarizing dichroism waveplate. The circular polarization dichroism ($\mathrm{CPD}=I_{RCP}-I_{LCP}$) in the transmission mode at a 2.4 μm wavelength reaches 83% and the extinction ratio is 20 dB. The excellent broadband performance of the quarter-wave plate is illustrated by the interaction of two resonance modes, which consist of an effective guided-mode resonance for the TE incident wave and hybrid gap-SPPs (G-SPPs) for the TM incident wave. These six polarization elements are assembled to form a full Stokes large pixel that almost accurately measures arbitrary polarized light in a broad range (1.2–2.8 μm).

## 2. Materials and Methods

In Figure 1a, the geometric parameters of the six proposed metasurfaces are specified, respectively. Pixels P2, P1, P3, P4, P5 and P6 represent the 0-degree, 90-degree, 135-degree and 45-degree polarizers, and the right and left circular dichroic devices, respectively. Pixel P2 can be rotated by 90°, 45°, and −45° to obtain pixels P1, P3, and P4. P5 can be transformed into P6 by performing a mirror-image transformation on the YZ cross-section. As shown in Figure 1b, pixel P5 is composed of a dual-major-axis plasmonic grating with the optical quarter-wave plate function, a silica support layer, and a 45° linearly polarized silver grating (P4). Figure 1c shows the period and width of the 0-degree polarizer. Figure 1d shows the top view of two metal gratings in pixel P5, and the angle θ is 45°. The refractive index of the SiO2 layer in the near-infrared band is 1.46, and the optical refractive index of silver is derived from reference [38]. The commercial software comsol multiphysics 5.6 is employed to investigate the optical properties of these metasurfaces. We impose the periodic boundary condition in the x- and y-directions Meanwhile, the perfectly matched layer (PML) and the waveguide port are used as the boundary condition in the z-axis direction, and the S-parameters of the transmitted light are extracted to obtain the corresponding amplitude and phase information.

## 3. Results

### 3.1. The Jones Matrix of the Device

In this section, we demonstrate full-Stokes-imaging polarimetry by analyzing the Jones matrix of the nanoparticle dimensions of Section 2. In addition, it is assumed that the Jones matrix of the polarization element is $\left[\begin{array}{cc} T_{xx} & T_{yx}\\ T_{xy} & T_{yy} \end{array}\right]$, which can be obtained by extracting the S-parameters. Figure 2a,c shows that $\left|T_{xx}\right|$ and $\left|T_{xy}\right|$ are equal, and $\left|T_{yx}\right|$ and $\left|T_{yy}\right|$ are equal. Figure 2b shows that arg($T_{xy}$)- arg($T_{xx}$), arg($T_{yx}$)- arg($T_{xx}$), and arg($T_{yy}$)- arg($T_{xx}$) of pixel P5 can all approximate the constant, which are 180 degrees, −90 degrees and 90 degrees, respectively. Meanwhile, arg($T_{xy}$)- arg($T_{xx}$), arg($T_{yx}$)- arg($T_{xx}$), and arg($T_{yy}$)- arg($T_{xx}$) of pixel P6 are 0 degrees, 90 degrees and 90 degrees, respectively, as shown in Figure 2d. From the four figures above, we can draw the conclusion that the Jones matrix of the circular dichroic devices, called T5 and T6, can be $\left[\begin{array}{cc} a\times i & b\\ -a\times i & -b \end{array}\right]$ and $\left[\begin{array}{cc} -a\times i & b\\ -a\times i & b \end{array}\right]$, respectively. Here, the elements of the matrix a and b are real numbers. In addition, T5×$\left[\begin{array}{c} 1\\ -i \end{array}\right]$ = $i\times \left[\begin{array}{c} a-b\\ b-a \end{array}\right]$, T5×$\left[\begin{array}{c} 1\\ i \end{array}\right]$ = $i\times \left[\begin{array}{c} a+b\\ -b-a \end{array}\right]$, T6×$\left[\begin{array}{c} 1\\ -i \end{array}\right]$ = $-i\times \left[\begin{array}{c} a+b\\ b+a \end{array}\right]$, and T6×$\left[\begin{array}{c} 1\\ i \end{array}\right]$ = $i\times \left[\begin{array}{c} b-a\\ b-a \end{array}\right]$, which indicate that the right (left) circular dichroic device can depress the transmission of the left (right) circularly polarized light when a and b are equal. In fact, there is still a small but not negligible gap between most of the data points in Figure 2b,d and the theoretical value, so it is important to further optimize the simulation so as to bring the phase difference closer to 90 degrees or −90 degrees. In addition, it is necessary to re-emphasize that the combination of the QWP and the linear polarizer can modulate circularly polarized light, and we separately study the effect of the QWP and the line polarizer on the phase difference in steps of 10% of the standard geometrical parameters in Figure 1, using the scanning methods with all the combinations. Figure 2e demonstrates the ability of the QWP to optimize the phase difference. Here, a1 is 81 nm, 90 nm and 99 nm, b1 is 360 nm, 400 nm and 440 nm, and h1 is 315 nm, 350 nm and 385 nm, and the number of parameter scans of the simulation model is 3 to the third power. The error E1 and E2 in Figure 2e,f are 1/3× $\int \left(\sqrt{{\left(q1\right)}^{2}+{\left(q2\right)}^{2}+{\left(q3\right)}^{2}}\right)d\lambda /\int d\lambda$ and 1/3* $\int \left(\sqrt{{\left(q4\right)}^{2}+{\left(q5\right)}^{2}+{\left(q6\right)}^{2}}\right)d\lambda /\int d\lambda$, respectively, where q1 is arg($T_{xy}$)- arg($T_{xx}$), q2 is arg($T_{yx}$)- arg($T_{xx}$), q3 is arg($T_{yy}$)- arg($T_{xx}$)−π/2, q4 is arg($T_{xy}$)- arg($T_{xx}$)−π, q5 is arg($T_{yx}$)- arg($T_{xx}$)+π/2, and q6 is arg($T_{yy}$)- arg($T_{xx}$)−π/2. The error E0 is (E1 + E2)/2. The serial number 14 corresponds to Figure 2b,d, which has an error of 6°. Meanwhile, the smallest error occurs in the case of sequence number 6, which is about 4°. However, the manipulation effect of the linear polarization grating in Figure 2f is significantly different from that of the QWP. It is very clear that the changes in the height and width of the linear polarizer grating have almost no effect on the phase difference. This is very exciting news, and it means that we may not need to be too concerned about its machining deviation during the actual process.

### 3.2. The Error Analysis of the Full Stokes Pixel

Due to the non-negligible ohmic loss of the plasmonic metasurface, the derivation formula of the corresponding Stokes matrix of the optical elements will change. The Jones matrix of the 0-degree polarizer can be expressed as $\left[\begin{array}{cc} p & 0\\ 0 & 0 \end{array}\right]$, and $\;M_{\theta }$ × $\left[\begin{array}{cc} p & 0\\ 0 & 0 \end{array}\right]$ × $M_{-\theta }$ represents the Stokes parameter of the θ-degree polarizer, where p is real number, $M_{\theta }\;$ is $\left[\begin{array}{cc} \cos\left(\theta \right) & \sin\left(\theta \right)\\ -\sin\left(\theta \right) & \cos\left(\theta \right) \end{array}\right]$, and $M_{\theta }\;$ is $\left[\begin{array}{cc} \cos\left(\theta \right) & -\sin\left(\theta \right)\\ \sin\left(\theta \right) & \cos\left(\theta \right) \end{array}\right]$. The components of the Stokes parameter are defined as:(1)$$S_{0}=(I_{0}+I_{90})/|p{|}^{2}$$
(2)$$S_{1}=(I_{0}-I_{90})/|p{|}^{2}$$
(3)$$S_{2}=(I_{135}-I_{45})/|p{|}^{2}$$
(4)$$S_{3}=(I_{rcp}-I_{lcp})/(4ab)$$

Here, $I_{0}$, $I_{90},\;I_{135}$, $I_{45},\;I_{rcp}$ and $I_{lcp}$ are the transmission intensities of the 0-degree, 90-degree, 135-degree and 45-degree polarizers, and the right and left circular dichroic devices, respectively. We apply the linearly polarized light represented by KK and Theta(θ)  to illuminate the six small pixels. Here, KK is the ratio of the electric field amplitude in the x- and y-directions, and θ is the phase difference between the electric field in the y-axis and x-axis directions. The intensities of the transmitted light obtained by integrating the power flux over the transmission port are input into Equations (1)–(4) to obtain the full Stokes parameter. In addition, a quantitative comparison by extracting the average errors of S1, S2 and S3 is important. The theoretical Stokes parameter is ($D_{0},D_{1},D_{2},D_{3}$), and the Stokes parameter obtained by the finite-element algorithm can be denoted by ($S_{0},S_{1},S_{2},S_{3}$). In addtion, errors for the degree of linear and circular polarizations are defined as 10×log($\left|\sqrt{S_{1}^{2}+S_{2}^{2}}/S_{0}-\sqrt{D_{1}^{2}+D_{2}^{2}}/D_{0}\right|)$ and 10×log($\left|S_{3}/S_{0}-D_{3}/D_{0}\right|)$, respectively. The errors of the degree of linear polarizations (Dolp) are less than −25 dB at the 1.2–2.8 μm wavelength range, and the average errors of the degree of circular polarizations (Docp) are less than −10 dB, as shown in Figure 3. Figure 3c,d also shows that the minimum error of the Docp is less than −30 dB, which corresponds to a circular dichroism extinction ratio greater than 15 dB.

## 4. Discussion

Figure 4a shows the transmission spectrum and phase difference spectrum of the quarter-wave plate (QWP) whose unit cell has two adjacent metal strips. There are three transmission peaks in the transmission spectrum of the QWP in the TM incident mode. The asymmetric transmission peak and the double-hook type of the phase difference spectrum at the 1.41 μm wavelength indicate the presence of Fano resonance in the device. In addition, two transmission peaks above the 50% average transmittance are shown in the TE incident mode. Meanwhile, the phase difference between the projection components of the transmitted beam in the two orthogonal polarization directions must satisfy the requirement of $270^{\circ }\pm 10^{{}^{\circ}}$ for an acceptable QWP. The light-blue area corresponds to the effective bandwidth (1.2 μm). Meanwhile, the device’s geometrical shape can be traced by the peculiar amplitude profile of the electromagnetic field. Figure 4b shows the field distribution of the QWP. For the TM incident mode, the optical near-field image, where the magnetic field leaks into the metal at both ends and the energy also accumulates in the dielectric layer, fits very well with the typical oscillatory characteristics of the gap-surface plasmon polaritons (G-SPPs) [39,40,41]. For the TE incident mode, the electric field energy is mainly concentrated in the left MIM waveguide with a narrow dielectric layer in the case of the 1.08 μm wavelength incidence, but the electric field energy is transferred to the right MIM waveguide at the 2.59 μm wavelength. The most likely explanation for the transmission peak is Fabry–Pérot (FP) resonance, and it should satisfy $2k_{z}h_{1}+2\Phi_{R}=2m\pi$, where m is an integer, $\Phi_{R}$ is the phase detected by the waveguide mode via the reflection at each of the openings, and $k_{z}$ corresponds to the phenomenological description in Figure 4b. Obviously, increasing the height of the grating to excite higher-order transmission peaks would be a suitable method to verify the effectiveness of the FP resonance. Figure 4c,d shows the relationship between the grating height and transmittance at some specific wavelengths, and the equidistant increase in the height of the grating means that the order of the transmission peak is increased by one. The field intensity distribution of the grating in the gray area from left to right represents the transmission peak with increasing order, and it mainly manifests as the increase in antinodes and nodes in standing waves. However, the field distribution map from right to left in the light-red area corresponds to the higher-order peaks. The phenomenon of the one-to-one correspondence between the height and transmission peak order confirms the dominant effect of the FP resonance on the QWP. Table 1 shows a comparison of the performance of the transmissive QWP appearing in several recent articles.

Figure 5a shows the transmission spectrum and extinction ratio spectrum of the 0-degree polarizer called pixel P2. The extinction ratio is $10\times \log\left(I_{0}/I_{90}\right)$, where $I_{0}$ and $I_{90}$ represent the transmittance in the case of the 0° and 90° polarized incidences, respectively. The average transmittance of the TM mode is greater than 80%, the average transmittance of the TE mode is close to 0, and the average extinction ratio is about 30 dB. As shown in Figure 5b, there is a strong energy localization in the air gap between the metal gratings in the TM mode, which manifests as a passband mode. However, there is almost no electric field present in the air gap channel in the TE mode, which manifests as a forbidden band mode. Figure 5c shows that the linear dichroism decreases as the incident angle decreases. Here, the linear dichroism is $I_{0}-I_{90}$. The peak of the linear dichroism decreases by half when the incident angle is 70°. In addition, it is very important to follow the reliability design of the device by clarifying the number of unit cells. A two-dimensional optical simulation model is used to calculate the transmittance of a finite-size grating. The computational domain of the model whose size increases with the increase in the number of grating cells is controlled by the lateral scattering boundary conditions and the perfectly matched layer in the vertical direction. The length of the probe that can detect transmittance is 30  μm. As shown in Figure 5d, the extinction ratio of the device increases with the increase in the number of cells, and the extinction ratio is approximately the theoretical value when the number of cells is 80.

In classical optics, a combination of a QWP and a linear polarizer with a special orientation can have a significant extinction effect on the right circularly polarized light. This idea is referenced to propose a bilayer plasmonic metasurface for constructing a circular dichroic device. Thick metal grating, a counterpart of the 45-degree polarizer, and dual-long-axis plasmonic grating, a counterpart of the QWP, make up the unit cells of the bilayer metasurface. In addition, the evaluation indexes of the circular polarization devices are generally the circular dichroism (CD) and the circular polarization extinction ratio (CDER). Here, the circular dichroism is $I_{rcp}-I_{lcp}$, the circular polarization extinction ratio is $10*\log\left(I_{rcp}/I_{lcp}\right)$, and $I_{rcp}$ and $I_{lcp}$ represent the transmittance in the cases of the right circularly polarized (RCP) and left circularly polarized (LCP) incidences, respectively. Figure 6a shows the performance of the right circular dichroic device named pixel P5, and the quality factor is defined as ∫(CD∗CDER)∗dλ/∫dλ. The maximum value of the quality factor corresponds to an 800 nm-thick silica support layer that is between the 45° linear polarizer and the QWP. As shown in Figure 6b, the transmittance of the circular dichroic device in the case of the LCP incidence is less than 0.1, and the average transmittance of the circular dichroic device in the case of the RCP incidence is greater than 60%. Meanwhile, the average extinction ratio at 1.1–2.8 μm of bandwidth can reach 12 dB, and the peak value at 2.2 μm can reach up to 22 dB. Figure 6c shows the electric field distribution of the device on the XZ and YZ cross-sections in the case of the 1.4μm circularly polarized incidence, and the biggest difference between the RCP and LCP occurs in the local electric field intensity in the silica support layer. The RCP can be transformed into the 45° linearly polarized light through a dual-long-axis grating that is a counterpart of the QWP, and then the 45° linearly polarized light proceeds unobstructed through the 45° linear polarizers into the air medium. However, the LCP transforms into the 135° linearly polarized light corresponding to the attenuation channel of the 45° linearly polarizers, and almost all the light energy is reflected back into the silicon medium below, resulting in a much stronger electric field remaining in the silica support layer. The field intensity plot at the 2.55 μm wavelength in Figure 6d is similar to that in Figure 6c, which also confirms the manipulation effect of the cascaded optical metasurface on the chirality of light. Table 2 shows a comparison of the performance of the circular polarizer appearing in several recent articles. The bandwidth should exceed an extinction ratio of 10 dB, and all values in Table 2 correspond to the simulated values.

## 5. Conclusions

In conclusion, we use a single-layer plasmonic metasurface consisting of a thick silver grating, an SiO_2_ support layer and silicon substrate to realize the function of 0-degree, 45-degree, 90-degree and 135-degree polarizers, and they all have an average extinction ratio of 30 dB during transmission at the λ=1 μm−3 μm operation bandwidth. In addition, a circularly polarizing dichroism device is proposed by using the bilayer plasmonic metasurface consisting of the dual-long-axis silver grating submerged in silicon substrate, the SiO_2_ support layer and the thick plasmonic grating, and the circular polarization dichroism ($\mathrm{CPD}=I_{RCP}-I_{LCP}$) at the 2.4 μm wavelength reaches 83% and the extinction ratio is 20 dB. We also numerically demonstratsoe that the full Stokes large pixel composed of six small pixels can almost accurately measure the arbitrary polarization state in the 1.2–2.8 μm wavelength range. The average errors of the degree of linear polarizations (Dolp) and degree of circular polarizations (Docp) are less than −25 dB and −10 dB, respectively. Such a low polarimetric error is likely to be useful in the field of biosensors and detection.

## Figures and Tables

**Figure 1 nanomaterials-13-02211-f001:**
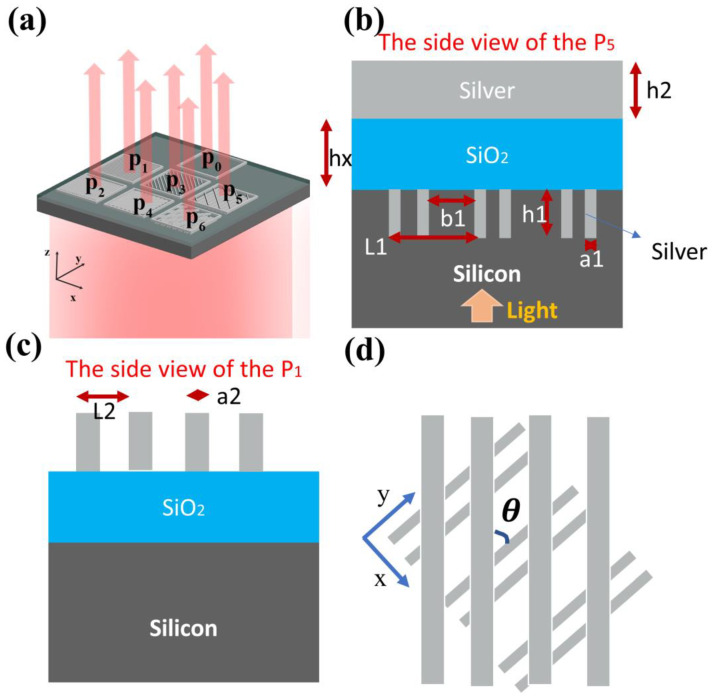
The 3-dimensional structure and side view of the device. (**a**) A schematic diagram of six small-pixel unit cells. P2, P1, P3, P4, P5 and P6 represent 0-degree, 90-degree, 45-degree and 135-degree polarizers, and left and right circular dichroic devices, respectively. The direction of the z-axis is the same as the direction of the beam propagation. (**b**) Side view of pixel P5 consisting of the dual-long-axis silver grating below, the silica support layer, and the silver grating above. L1 = 700 nm, a1 = 90 nm, h1 = 350 nm, b1 = 400 nm, hx = 800 nm, and h2 = 400 nm. The incident light is emitted from the silicon substrate below to the metal grating. (**c**) Side view of pixel P2. L2 = 495 nm; a2 = 200 nm. (**d**) Top view of two metal gratings in pixel P5, and the angle θ is 45. For the linear polarization gratings, the TM mode specifies that the electric field is perpendicular to the grating direction, while the TE mode specifies that it is parallel to the grating direction. The x-axis, y-axis, and z-axis form a spatial coordinate system that conforms to a right-handed spiral law coordinate system.

**Figure 2 nanomaterials-13-02211-f002:**
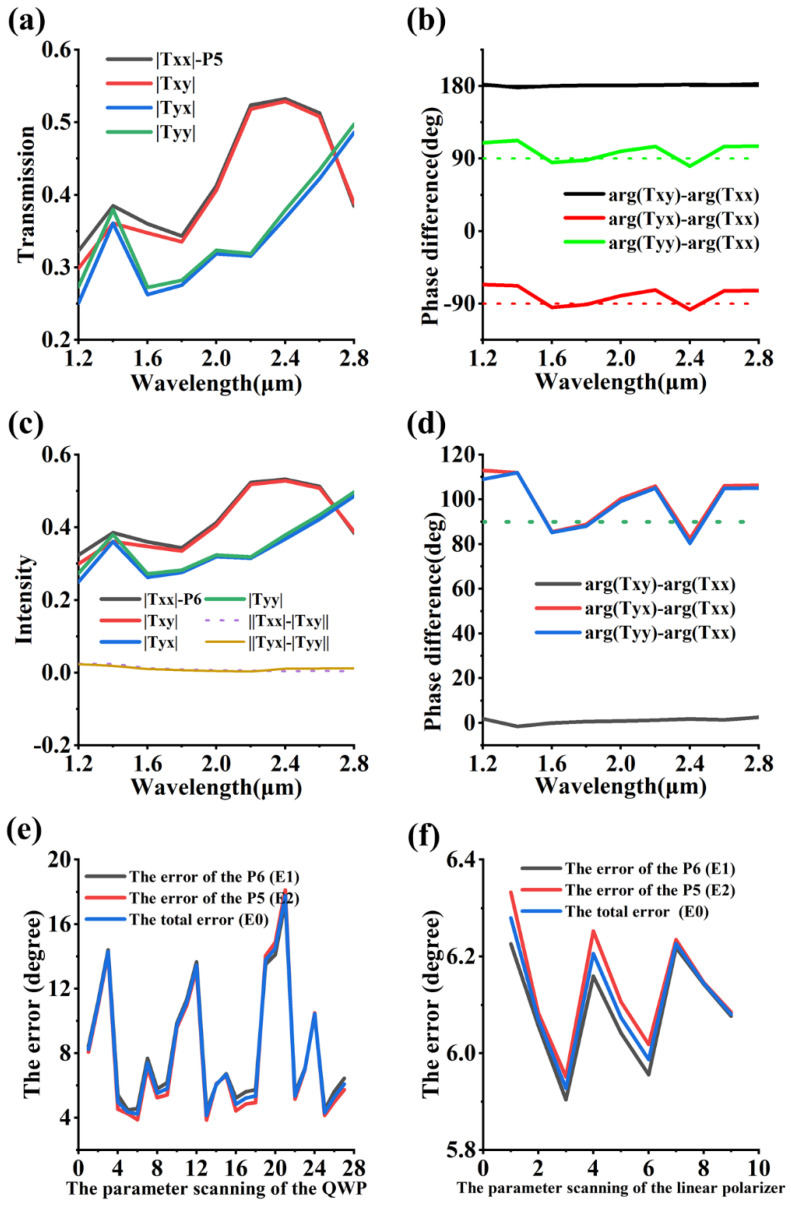
The transmission and phase difference spectrum. (**a**,**b**) Pixel P5. (**c**,**d**) Pixel P6. (**e**,**f**) the optimization process of the parameter scanning.

**Figure 3 nanomaterials-13-02211-f003:**
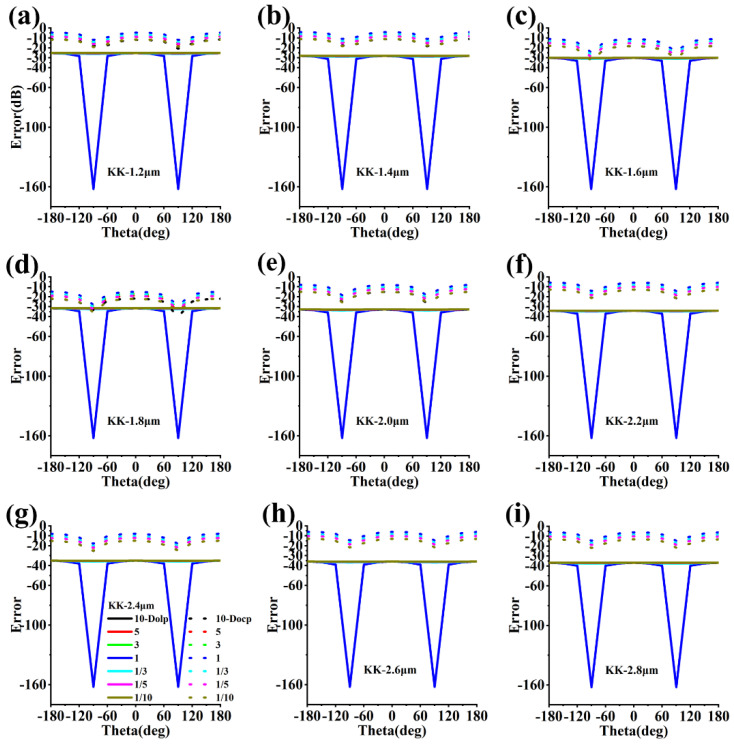
The error for degree of linear and circular polarizations. KK is the ratio of the electric field amplitude of the incident light in the x- and y-directions. KK: 10, 5, 3, 1, 1/3, 1/5, and 1/10. The solid line represents Dolp and the dotted line represents Docp. (**a**) The effect of different KK and θ on the error of polarization at 1.2 µm wavelength.; (**b**) The effect of different KK and θ on the error of polarization at 1.4 µm wavelength; (**c**) The effect of different KK and θ on the error of polarization at 1.6 µm wavelength; (**d**) The effect of different KK and θ on the error of polarization at 1.8 µm wavelength; (**e**) The effect of different KK and θ on the error of polarization at 2.0 µm wavelength; (**f**) The effect of different KK and θ on the error of polarization at 2.2 µm wavelength; (**g**) The effect of different KK and θ on the error of polarization at 2.4 µm wavelength; (**h**) The effect of different KK and θ on the error of polarization at 2.6 µm wavelength; (**i**) The effect of different KK and θ on the error of polarization at 2.8 µm wavelength.

**Figure 4 nanomaterials-13-02211-f004:**
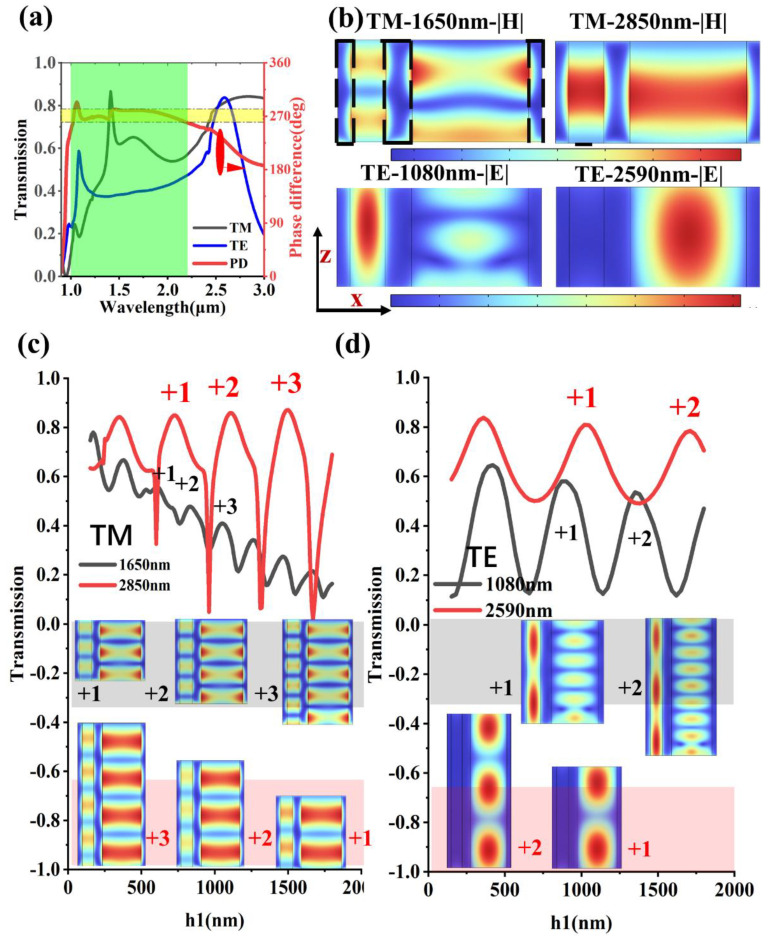
The transmission spectrum and mode analysis. (**a**) The transmission spectrum and phase difference spectrum of the QWP. The yellow area corresponds to a phase difference of 270 ± 10°, and the light-blue area corresponds to the effective bandwidth of the QWP. (**b**) The near-field distribution of the QWP in TE and TM incident modes. The corresponding part of the black dashed box is metal. (**c**,**d**) The functional relationship between transmittance and grating thickness, as well as the field distribution maps corresponding to the transmission peaks. The wavelength of red (gray) corresponds to the area of red (gray). The numbers with plus signs represent the corresponding transmission peaks.

**Figure 5 nanomaterials-13-02211-f005:**
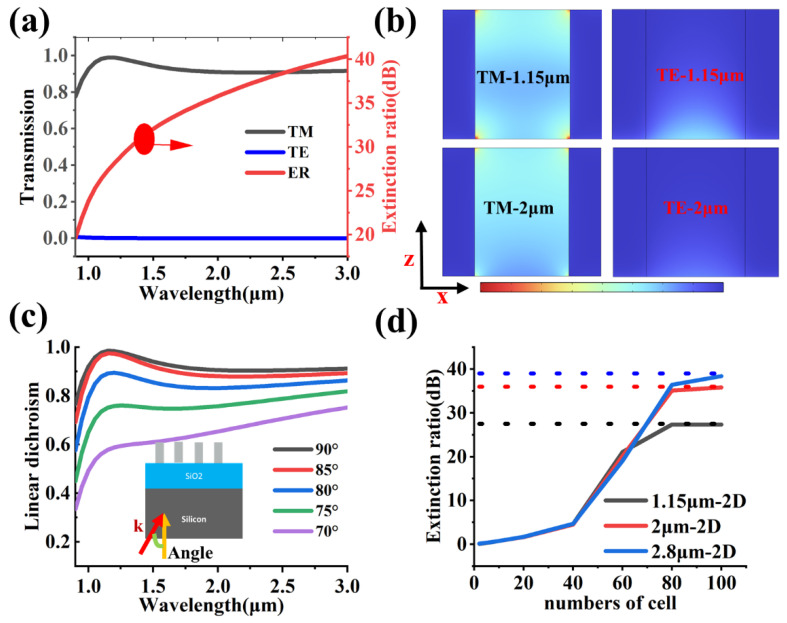
The performance of pixel P2. (**a**) The transmission spectrum and extinction ratio spectrum of pixel P2. (**b**) The electric field distribution of the linear polarization grating. (**c**) The functional relationship between the linear dichroism spectrum and the incident angle. The yellow arrow is the positive direction of the z-axis, and k is the direction of the incident light. (**d**) The relationship between the number of unit cells of the metasurface and the extinction ratio. The dashed line represents the ideal value of the extinction ratio. The TM incidence corresponds to 0-degree polarized light.

**Figure 6 nanomaterials-13-02211-f006:**
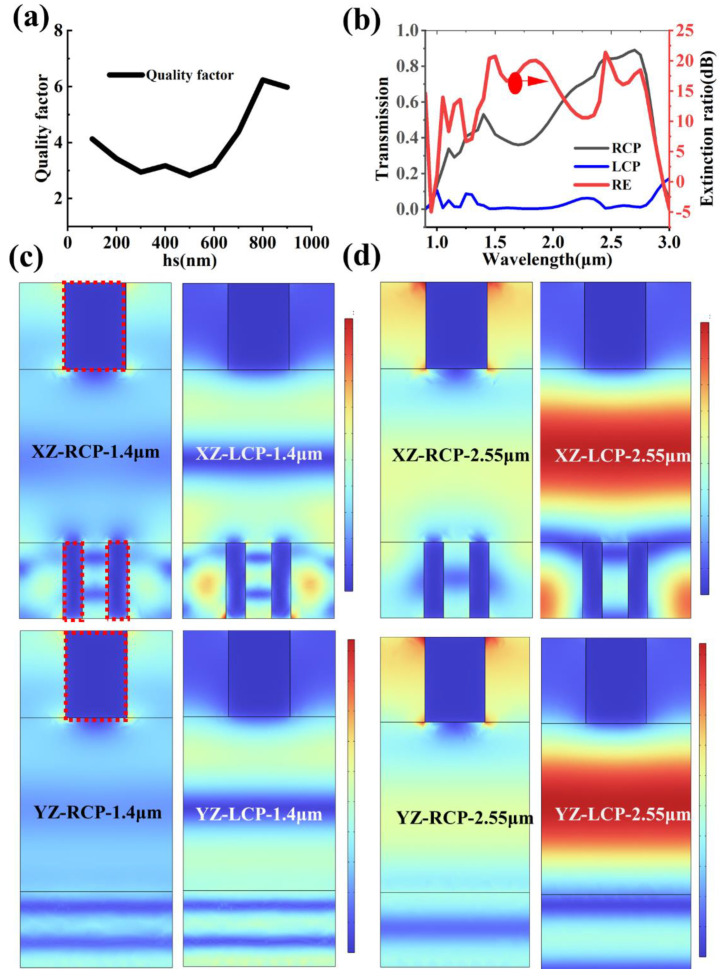
The performance of the circular polarizer. (**a**) The relationship between the thickness of silica support layer and the quality factor. (**b**) The transmission spectrum and extinction ratio spectrum of pixel P5, and the red dashed box represents the boundary of the silver. P_5_ (**c**,**d**): The cross-section electric field distribution of the circular dichroic metasurface.

**Table 1 nanomaterials-13-02211-t001:** Comparison of the characteristics of the QWP in the near-infrared band.

Structure Design	Wavelength	Efficiency (Average)	Bandwidth
L-shaped [42]	1550 nm	~0.4	80 nm
Metal grating [43]	1450 nm	~0.4	300 nm
Rectangular hole [44]	1500 nm	~0.4	~520 nm
Metal [45]	1550 nm	~0.4	120 nm
Metal cross [46]	1250 nm	0.85	~140 nm
Ag grating [39]	1250 nm	~0.7	600 nm
This proposal	1600 nm	~0.55	1200 nm

**Table 2 nanomaterials-13-02211-t002:** The performance of the circular polarizer.

Structure Design	Operation Mode	Bandwidth	Extinction Ratio (Maximum)	Circular Polarization Dichroism (Average)
Plasmonic metasurface [3]	Absorption	1.33–1.4 μm (70 nm)	~9:1	90%
Dielectric metasurface [22]	Transmission	1.48–1.52 μm (40 nm)	345:1	80%
Bilayer metasurface [47]	Transmission	0.75–0.85 μm (100 nm)	NA	8%
Metal–dielectric hybrid [23]	Transmission	1.42–1.60 μm (180 nm)	~400:1	~80%
Bilayer metasurface [24]	Transmission	3.7–3.9 μm (200 nm)	~300:1	9%
U-shaped metasurface [25]	Transmission	1.50–1.70 μm (200 nm)	~1900:1	~80%
This proposal:	Transmission	1.20–2.80 μm (1600 nm)	~100:1	~60%

## Data Availability

The data that support the findings of this study are available from the corresponding author upon reasonable request.

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
