# Peer review of "The Ultra-Large-Bandwidth Cascade Full-Stokes-Imaging Metasurface Based on the Dual-Major-Axis Circular Dichroism Grating"

_nanomaterials, 2023, doi:10.3390/nano13152211_

Round 1

Reviewer 1 Report

In this paper, Bo Cheng and Guofeng Song reported a cascade Full-Stokes imaging metasurface with an ultra-large bandwidth. Compared to other reported imaging metasurface, the proposed metasurface has a bandwidth several times broader. I found this study interesting and the authors’ discussion convincing. I would like to recommend accepting it for publication after some minor improvements:

(1) For sample schematics and simulated near-field distribution images, could the authors add the labels of the XYZ axis in order to better illustrate the sample configuration and let the reader easier to correspond near field with the sample configuration? For schematics in Figure 1, I would like to suggest the authors use silver color to indicate the silver film and structures instead of golden color.

(2) Authors claimed that arg(???)- arg(???), and arg(???)-arg(???) for P5 are 90 degree and -90 degree and for P6 are 90 degree and 90 degree. However, the results in Figure 2b and d illustrate that most data points are larger than 90 degree or -90 degree. Can the authors further optimize the simulation to most data points closer to 90 degree or -90 degree?

(3) In Figures 4 c and d. I think using numbers or alphabets to label the transmission peaks instead of triangles and ellipses is better.

Reviewer 2 Report

The auhors presente a manuscript describing their project of a large bandwidth SPP-based metamaterial for both linear and circular dichorism.

The work inlcudes the spectral analysis abouth the phase and amplitude of the filtered fields, together with a Full-Stokes Imaging Polarimetry; the quantitative evaluation has been performed numerically within the Comsol Multiphysics simulation environment.

The analysis is effectively complete and exhaustive in all its aspects, and the results appear consistent and reproducible by using ordinary numerical methods; however, this work lacks some originality, since the studied metastructures constitute just one of the possible variants of those already explored and widely presented in the rich literature of the sector. It is not necessary to indicate other examples of this same category in this instance, since the authors have already wisely shown some structures similar to the one they are presenting for efficiency comparisons, as shown at the end of chapter 4.

For this reason, I consider this work to be publishable although I reccomend the editors and the reader to merely consider it as a simple but clear example to design filtering metasurface.
Other than this, I suggest the authors just to perform very small optimizations of the English, and make the sense of some statements clearer/more suitable to the meaning they were aiming at (for example, please recast the phrase "The physical image of the device is clearly caused by the joint action" in Line 153 with a clearer maning - probably the authors meant to state that the device geometrical shape can be traced by the peculiar amplitude profile of the e.m. field).
I also suggest to slightly improve the reference list, adding some mentions regarding high-dichroic chiral structures (current references 15-19), for instance inspecting the work of Esposito et al.; here it follows a list with some worthy papers to mention:
https://www.nature.com/articles/s41598-017-05193-4
https://www.nature.com/articles/ncomms7484
https://onlinelibrary.wiley.com/doi/full/10.1002/adfm.202109258

I just suggest a brief chek on the quality of English, correcting some conjunctions and prepositions in few instances.

Reviewer 3 Report

The Manuscript describe about the ultra large bandwidth cascade Full stockes metasurface. Below are my comments:

Which TE mode please? 01 or 02?

If you have performed any error analysis then specify over abstract section.

What is the extension of circular polarized wave?

Lamda in line 60 signifies the fundamental wavelength?? Specify it.

What are the values of relative dielectric constant and loss tangent of your proposed surface?

Line 139 to 140 could be placed in conclusion section.

Provide the experimental results too.

Overall, the manuscript required the experimental analysis. Better to include these aspects.

Fine changes are required. 

Round 2

Reviewer 3 Report

The overall changes are satisfactory. The manuscript presents the simulated results and I am doubtful as there is no any experimental analysis. 

Minor correction needed. 
